# Impact of developmental coordination disorder in childhood on educational outcomes in adulthood among neonatal intensive care recipients: a register-based longitudinal cohort study

Isak Persson,[1] Filipa Sampaio ![ORCID],[2] Tengiz Samkharadze,[2] Richard Ssegonja ![ORCID],[2,3] Kine Johansen ![ORCID] [1]

¹Department of Women's and Children's Health, Uppsala University, Uppsala, Sweden
²Department of Public Health and Caring Sciences, Uppsala University, Uppsala, Sweden
³Department of Medical Sciences, Uppsala University Hospital, Uppsala, Sweden

**Correspondence to**
Dr Kine Johansen;
kine.johansen@kbh.uu.se

## ABSTRACT

**Objectives** Developmental coordination disorder (DCD) is related to poorer educational outcomes among children and adolescents. Evidence on this association into adulthood is lacking. Therefore, we aimed to investigate whether probable DCD (pDCD) in childhood affected educational outcomes among adults, and whether this was affected by sex or a co-occurring attention deficit in childhood.

**Design** Register-based longitudinal cohort study.

**Setting** Neonatal intensive care (NIC) recipients born at Uppsala University Children's Hospital, Uppsala, Sweden, from 1986 to 1989 until they reached the age of 28.

**Participants** 185 NIC recipients.

**Primary and secondary outcome measures** At the age of 6.5, 46 (24.6%) of the NIC recipients were diagnosed with pDCD. Using register-based longitudinal data, we compared participants with and without pDCD in terms of: (1) age at Upper Secondary School (USS) graduation, and (2) highest level of education achieved by age 28.

**Results** The median age at USS graduation was 19 years, with similar graduation ages and ranges between those with or without pDCD. However, a higher proportion of participants without pDCD had graduated from USS at ages 19 and 24. By age 29, most participants had completed USS. At age 28, 33% of participants had attained a bachelor's or master's degree. Although there was no significant difference between the groups, the proportion that had attained a degree was higher among those without pDCD and women without pDCD had achieved the highest level of education. Educational outcomes remained similar for those with pDCD, regardless of childhood attention deficit.

**Conclusions** pDCD during childhood may have a lasting impact on educational outcomes, particularly among women. Raising awareness of DCD among parents, health and educational professionals is vital for early identification and the provision of appropriate support and interventions in schools, mitigating the potential negative consequences associated with DCD and promoting positive educational outcomes.

## STRENGTHS AND LIMITATIONS OF THIS STUDY

⇒ Strengths of this study include the longitudinal design and the use of national registers with high coverage and a low rate of missing data.

⇒ It is one of few papers that follows educational outcomes in children with probable developmental coordination disorder into adulthood, and the only one of these focusing on developmental coordination disorder specifically.

⇒ Educational outcomes were measured solely as a binary outcome, without considering academic performance or any extra support received to graduate.

⇒ Our analyses would have benefitted from a comparison to a population-matched control group and measures of participants' functioning in adulthood for validation.

⇒ The special characteristics of the study sample and the small sample size, could potentially limit the generalisability of the results and the ability to detect true differences, especially in subgroup analyses.

## INTRODUCTION

Developmental coordination disorder (DCD) is a neurodevelopmental disorder that affects motor skills, which significantly interferes with daily activities, such as play, school-related and leisure activities.[1] With a prevalence of 5%–6% among children,[1] DCD affects approximately 1 in every 20 school-aged children. Moreover, it frequently co-occurs with other neurodevelopmental disorders, such as attention deficit hyperactivity disorder (ADHD), dyslexia, dyscalculia, developmental language disorder and autism spectrum disorder.[1 2] While DCD once was considered a childhood disorder, research suggests that it is a lifelong condition.[1] Knowing that DCD affects school-related

activities it is important to know if the disorder affects educational outcomes, as this closely correlates with both future employment opportunities and higher incomes.[3]

Children with DCD are more likely to have lower educational outcomes in childhood and adolescence,[4–7] experience difficulties with language skills,[8] impaired numerical cognition abilities,[9] and executive dysfunction compared with typically developing peers.[1 9] Furthermore, DCD is also known to affect handwriting,[1] which can hinder academic success. Harrowell et al reported lower national examinations grades for 16-year-old English adolescents with DCD compared with their peers without DCD.[5] In addition, De Waal et al found that girls with DCD achieved better outcomes than boys with DCD.[10]

However, educational outcomes in children with DCD can be affected by several other factors. For example, prematurity and ADHD are well-known to be associated with poorer educational outcomes in childhood and adolescence.[11–14] As premature born children are at increased risk of neurodevelopmental disorders,[15] and that neurodevelopmental disorders commonly co-occur, further complicates the picture.[2] In fact, studies suggest that individuals with ADHD and a co-occurring DCD tend to have poorer educational outcomes than those diagnosed with ADHD alone.[6 7 16] Other factors, such as socioeconomic background, parental educational level, migration background, and income can also affect educational outcomes.[17] Moreover, DCD may affect boys and girls differently. For example, a study by Brown and Cairney found that girls with poor motor coordination had lower self-perceptions of academic competence, athletic ability and physical appearance compared with boys with DCD,[18] and these differences became more pronounced with age.

In Sweden, education is tax-financed and offered free-of-charge. After 9 years of compulsory schooling, that is, at approximately 15 years of age, the majority of the adolescents choose to start Upper Secondary School (USS), an elective preparatory or vocational programme lasting for 3 years.[19] Graduation from USS is a prerequisite for all higher education and is commonly requested in the labour market for employment, even for occupations where educational requirements have traditionally been low.[20] In 2021, 31.4% of the Swedish labour force without graduation from USS were unemployed, compared with 7.8% among those who had graduated from USS without completing any post-secondary education.[21] According to the Delegation for the Employment of Young People and Newly Arrived Migrants, 65% of all students born in 1989 graduated from USS before age 20 and 91% before age 25.[22] There was also a considerable gender difference; at 25 years of age, 12% of men lacked graduation compared with 6% of women.[22] The societal cost of one individual not finishing USS in Sweden is estimated at €163 000.[23] In 2021, 30% of the population aged 25–64 had attained at least a bachelor's degree,[24] and the average age to reach this level was 28 years.[3] At age 40, 39% of women and 26% of men had attained at least a bachelor's degree.[25]

Although poorer educational outcomes for individuals with DCD is well-documented in childhood and adolescence, there is a paucity of studies examining if this association persists into adulthood. Therefore, using register-based data from a cohort of Swedish neonatal intensive care (NIC) recipients, we explored, how meeting the diagnostic criteria for DCD in childhood affected educational outcomes in adulthood. Additionally, we investigated whether sex or a co-occurring attention deficit in childhood affected educational outcomes.

## MATERIAL AND METHODS
### Study design and participants
In this register-based longitudinal cohort study, we used data from a cohort of NIC recipients born at Uppsala University Children's Hospital, Sweden, between 1986 and 1989. All surviving infants (n=246) treated at the NIC unit during this period, who were residents in Uppsala County, were enrolled in a longitudinal follow-up study.[26] Inclusion criteria were: (1) ventilator treatment, (2) treatment with continuous positive airway pressure (CPAP), (3) perinatal asphyxia (Apgar score ≤4 at 5 min), (4) neonatal convulsions treated with continuous intravenous anticonvulsive drugs, (5) need for total parenteral nutrition in the neonatal period, and (6) all very and extremely preterm born infants (<32 completed gestational weeks (GW)) irrespectively of any of the other criteria.[26]

The children participated in repeated assessments from estimated date of birth up to 10 years of age.[27 28] At 6.5 years, 212 children were assessed using the Standardised Test of Motor Impairment (TOMI),[29] the predecessor of the Movement ABC,[30] the Motor-Perceptual Development (MPU), 0-7 years,[31] and the Combined Assessment of Motor Performance and Behaviour (CAMPB).[27 32] TOMI evaluates children's motor abilities, specifically manual dexterity, ball skills, and balance,[29] while MPU assesses gross motor skills, eye-hand coordination, perception, activities of daily living, and academic skills.[31] The CAMPB model,[32] developed during the initial cohort study, was used to evaluate the children's coordination and attention.[32] Based on this assessment, the children were retrospectively classified into motor deviation categories,[27] including DCD defined according to the Diagnostic and Statistical Manual of Mental Disorders Text Revision Fourth Edition (DSM-IV-TR)[33] (n=46). For the purpose of the present study, those diagnosed with cerebral palsy (n=20) and mental retardation (n=7) were excluded. This left us with 185 participants eligible for participation (online supplemental file 1). During the initial study period, Uppsala County had 280 000 inhabitants with a slightly younger and slightly more highly educated population than the national average.[28]

This study adhered to the Strengthening the Reporting of Observational Studies in Epidemiology cohort checklist recommendations.[34]

## Data sources

We collected data on sex, gestational age, and outcome of the motor assessment at 6.5 years from the initial cohort study. In 2019, data on educational outcomes was retrieved from the 'Longitudinal Integrated Database for Health Insurance and Labour Market Studies' (LISA)[35] and linked to data on diagnoses and healthcare consumption from the 'National Patient Register'[36] and the 'Cause of Death Register'.[37] Linkage of registers was possible due to the personal identity numbers allocated to all Swedish residents. The National Board of Health and Welfare matched and anonymised the data before delivering it to the last author. Data were available for the years 2002–2018.

## Outcome variables

The LISA-register integrates yearly information about education, employment and demographic characteristics. Education level is reported on an ordinal scale ranging from less than 9 years of compulsory school to a licentiate's or doctor's degree.[38] Using this variable, we derived several variables, including a continuous variable indicating the age at which participants graduated from USS based on their reported year of graduation. We also created three categorical variables, determining whether participants had graduated from USS by age 19, 24 and 29 (yes/no). The age-specific binary variables for graduation from USS at 19 and 24 years of age were chosen based on data on average graduation rates in the Swedish population born in 1989,[22] while 29 years was the final point of measurement that allowed for analysis of the entire sample. Additionally, we generated a binary variable to indicate whether participants had graduated from USS by the end of the study. Finally, we created a categorical variable indicating their educational level at age 28, based on the avarage age for attaining a bachelor's degree in Sweden, which is 28 years.[3]

## Measures

The participants were categorised into two groups based on whether or not they met the diagnostic criteria for *DCD* at 6.5 years of age[27]: that is, those without motor deviations, as well as those assessed as having motor delays or motor problems, were grouped together as 'No DCD', and compared with those who met the DCD criteria according to DSM-IV-TR, named 'probable DCD' (pDCD). *Sex* was analysed as a binary measure (men/women). Participants assessed with a minor or pronounced attention deficit according to CAMPB at 6.5 years were defined as having a co-occurring attention deficit in childhood.[27] *Gestational age* was categorised as extremely preterm (22–27 GW), very preterm (28–32 GW), moderately preterm (32–36 GW), or full-term (37–42 GW). Diagnose codes obtained from the 'National Patient Register' were grouped according to the International Statistical Classification of Diseases and Related Health Problems-eighth/ninth revision (ICD-8/9), but not specified in detail. Therefore, all information about diagnoses of neurodevelopmental disorders were combined into one variable indicating whether or not participants had any *neurodevelopmental disorder* (yes/no). We categorised the diagnosis groups recorded during participants' first year of life as either posing an increased risk of motor problems or not. The diagnoses that posed an increased risk were summed into a continuous variable representing the total *number of diagnoses*. See online supplemental files 2 and 3 for a detailed list of ICD-codes. *Hospital days* describe the total number of days the participants spent in inpatient care during their first year of life.

## Missing data

Two women with pDCD missed all educational data in the LISA-register and were excluded from all statistical analyses, leaving a total sample of 183 participants. Additionally, two other women with pDCD had incomplete register data. Data from these participants were used when values could be assigned to the categorical outcome variables indicating graduation from USS, but were otherwise excluded. When analysing level of education at 28 years of age, four participants who had emigrated were lost to follow-up.

Nine participants had missing data on diagnoses and hospital days during the first year of life. To minimise loss of data when performing the regression analysis, we imputed the data using multiple imputation by chained equations. Five datasets were imputed using all available variables as predictors.

## Statistical analysis

Descriptive statistics were used to characterise our sample and are presented as numbers and percentages. We compared participants with or without pDCD at 6.5 years of age. Furthermore, subgroup analyses were performed to investigate educational outcomes based on sex and, among participants with pDCD, the presence of a co-occurring attention deficit during childhood. Differences in group characteristics were analysed using the Mann-Whitney U test, $\chi^2$ test or Fischer's exact test. Effect sizes were calculated using Hedges g and Cohen's d, and should be interpreted as small ($\geq 0.20$), medium ($\geq 0.50$) and large ($\geq 0.80$) effects.[39] Logistic regressions were conducted to explore the relationship between pDCD at 6.5 years of age (categorical) and having graduated from USS at 19, 24 or 29 years of age (categorical). The model was adjusted for sex (categorical), gestational age (categorical), any diagnosis of a neurodevelopmental disorder (categorical), the number of diagnoses posing a risk of motor problems (continuous), and hospital days during the first year of life (continuous). A two-tailed p value of <0.05 was considered statistically significant. Effect sizes were calculated using Social Science Statistics (https://www.socscistatistics.com/effectsize/default3.aspx), while all other analyses were performed using IBM SPSS Statistics V.28 for Windows.

## Patient and public involvement

No patient or public involvement.

**Table 1** Sample characteristics

|  | Total | No DCD | pDCD |  |  |
|---|---|---|---|---|---|
|  | n (%) | n (%) | n (%) |  |  |
|  | 183 (100) | 138 (75.4) | 45 (24.6) | P value | g |
| **Sex** |  |  |  |  |  |
| Men | 108 (59.0) | 80 (58.0) | 28 (62.2) | 0.615 |  |
| Women | 75 (41.0) | 58 (42.0) | 17 (37.8) |  |  |
| **Gestational age** |  |  |  |  |  |
| 22–27 GW | 9 (4.9) | 5 (3.6) | 4 (8.9) | 0.502 |  |
| 28–31 GW | 42 (23.0) | 31 (22.5) | 11 (24.4) |  |  |
| 32–36 GW | 70 (38.3) | 55 (39.9) | 15 (33.3) |  |  |
| 37–42 GW | 62 (33.9) | 47 (34.1) | 15 (33.3) |  |  |
| **Neurodevelopmental disorder** |  |  |  |  |  |
| Yes | 14 (7.7) | 6 (4.3) | 8 (17.8) | 0.007 |  |
| No | 169 (92.3) | 132 (95.7) | 37 (82.2) |  |  |
| **Number of diagnoses posing a risk of motor problems** |  |  |  |  |  |
| Median | 4.0 | 4.0 | 4.0 | 0.976 |  |
| Min–max | 0.0–10.0 | 0.0–10.0 | 1.0–9.0 |  |  |
| IQR | 2.0–5.0 | 2.0–5.0 | 2.0–5.0 |  |  |
| Mean (SD) | 4.0 (1.9) | 4.0 (1.9) | 4.0 (1.9) |  | 0.00 |
| Missing | 9.0 (4.9) | 7.0 (5.1) | 2.0 (4.4) |  |  |
| **Number of hospital days** |  |  |  |  |  |
| Median | 24.0 | 23.5 | 26.0 | 0.643 |  |
| Median | 22.0 | 22.0 | 24.0 |  |  |
| Min–max | 0.0–200.0 | 1.0–200.0 | 0.0–168.0 |  |  |
| IQR | 8.8–45.0 | 8.0–43.0 | 9.0–53.0 |  |  |
| Mean (SD) | 32.1 (30.5) | 30.8 (29.2) | 35.9 (34.1) |  | 0.17 |
| Missing | 9 (4.9) | 7 (5.1) | 2 (4.4) |  |  |

DCD, developmental coordination disorder; GW, gestational weeks; pDCD, probable DCD.

## RESULTS

### Sample characteristics

Of the 183 participants, slightly more than half were men (n=108, 59.0%) (table 1). The majority were born moderate to late preterm (n=70, 38.3%) or full-term (n=62, 33.9%). The corresponding numbers for the very and extremely preterm were 42 (23.0%) and 9 (4.9%), respectively. In total, 14 participants (7.7%) had a registered neurodevelopmental disorder recorded in specialised outpatient care (online supplemental file 3). Of these, 8 (57.1%) were assessed with pDCD at 6.5 years of age. The study sample had a median of four diagnoses that posed a risk of motor problems (IQR 2.0–5.0) recorded during their first year of life, and they spent a median of 24 days in hospital during the same period (IQR 8.8–45.0).

Except that participants with pDCD had a higher rate of neurodevelopmental disorder in specialised outpatient care than those without pDCD (p=0.007), no other statistically significant group differences were found in sample characteristics. The pDCD group contained slightly more men (n=28, 62.2%) and had a median of 24 hospital days during their first year of life, which was 2 days longer than the median stay of 22 days for the No DCD group. When comparing the distribution between gestational age groups, it can be noted that the pDCD group had a higher relative count of extremely preterm births (n=4, 8.9%) compared with those without pDCD (n=5, 3.6%), although not statistically significant (p=0.502).

### Educational outcomes

The median age at USS graduation for all participants was 19 years (table 2). While those with pDCD graduated with a slightly wider spread (IQR 19.0–20.0 vs 19.0–19.0) (p=0.022, g=0.41), the graduation age and range were similar between the groups. At 19 years, 78.2% (n=108) of those without pDCD and 54.5% (n=24) of those with pDCD had graduated from USS (p=0.002). At 24 years, the percentage of graduates remained higher among those without pDCD (92.8%, n=128) than among those with pDCD (81.1%, n=36) (p=0.044).

**Table 2** Educational outcomes for participants with or without pDCD at 6.5 years of age

| | Total | No DCD | pDCD | | |
|---|---|---|---|---|---|
| | n (%) | n (%) | n (%) | | |
| | 183 (100) | 138 (75.4) | 45 (24.6) | P value | g |
| Age (years) at graduation from USS | | | | | |
| Median | 19.0 | 19.0 | 19.0 | 0.022 | |
| Min–max | 19.0–26.0 | 19.0–26.0 | 19.0–26.0 | | |
| IQR | 19.0–19.0 | 19.0–19.0 | 19.0–20.0 | | |
| Mean (SD) | 19.3 (1.0) | 19.2 (0.8) | 19.6 (1.4) | | 0.41 |
| Missing data, total | 18 (9.8) | 9 (6.5) | 9 (20.0) | | |
| No graduation | 16 (8.7) | 9 (6.5) | 7 (15.6) | 0.069 | |
| Absent/incomplete register data | 2 (1.1) | | 2 (4.4) | | |
| USS graduation at 19 years | | | | | |
| Yes | 132 (72.5) | 108 (78.3) | 24 (54.5) | 0.002 | |
| No | 50 (27.5) | 30 (21.7) | 20 (45.5) | | |
| Missing data | 1 (0.5) | | 1 (2.2) | | |
| USS graduation at 24 years | | | | | |
| Yes | 164 (90.1) | 128 (92.8) | 36 (81.8) | 0.044 | |
| No | 18 (9.9) | 10 (7.2) | 8 (18.2) | | |
| Missing data | 1 (0.5) | | 1 (2.2) | | |
| USS graduation at 29 years | | | | | |
| Yes | 166 (91.2) | 129 (93.5) | 37 (84.1) | 0.069 | |
| No | 16 (8.8) | 9 (6.5) | 7 (15.9) | | |
| Missing data | 1 (0.5) | | 1 (2.2) | | |
| Level of education at 28 years | | | | | |
| Compulsory school, 9 years | 10 (5.6) | 7 (5.1) | 3 (7.3) | 0.080 | |
| Upper secondary education, ≤2 years | 6 (3.4) | 2 (1.5) | 4 (9.8) | | |
| Upper secondary education, 3 years | 74 (41.6) | 55 (40.1) | 19 (46.3) | | |
| Post-secondary education, <3 years | 29 (16.3) | 24 (17.5) | 5 (12.2) | | |
| Bachelor's or master's degree | 59 (33.1) | 49 (35.8) | 10 (24.4) | | |
| Missing data | 5 (2.7) | 1 (0.7) | 4 (8.9) | | |

DCD, developmental coordination disorder; pDCD, probable DCD; USS, Upper Secondary School.

In total, 8.7% (n=16) had not finished USS by the end of the study, seven of whom (43.8%) were assessed with pDCD at 6.5 years of age (p=0.069). Regardless, the majority of participants (91.2%) had graduated from USS at 29 years. At 28 years, 33% of participants had attained a bachelor's or master's degree, with no statistically significant difference between the groups (p=0.080). However, a higher proportion of participants without pDCD (35.8%) had attained a bachelor's or master's degree compared to those with pDCD (24.4%).

The contribution of covariates in explaining changes in graduation from USS was small (table 3). However, a pDCD diagnosis negatively impacted graduation at both 19 and 24 years of age, in both unadjusted and adjusted models. Those with pDCD at 6.5 years of age were less likely to graduate at 19 (OR: 0.333, p=0.003; 95% CI 0.16 to 0.68) and 24 (OR: 0.352, p=0.041; 95% CI 0.13 to 0.96) years of age, compared with those without pDCD. Adjustment for sex, gestational age, any neurodevelopmental disorder, number of diagnoses posing a risk of motor problems, and the number of hospital days during the first year of life did not change the odds of graduation significantly (OR: 0.315–0.345).

### Educational outcomes by sex

There were no statistically significant differences in educational outcomes between men and women in the total sample, nor when comparing participants with or without pDCD, respectively (online supplemental file 4). However, women with pDCD (n=8, 50.0%) had a significantly lower proportion of USS graduates at age 19 compared with women without pDCD (n=48, 82.8%) (p=0.017, g=0.85) (table 4). While not statistically significant, there was a similar tendency among men (p=0.075,

**Table 3** Logistic regression results for prediction of graduation at 19, 24 and 29 years of age

| Covariates | $R^2$ | B | OR (95% CI) | P value |
|---|---|---|---|---|
| **USS graduation at 19 years** | | | | |
| Unadjusted | 0.069 | | | |
| pDCD (pDCD vs No DCD) | | 1.099 | 0.333 (0.16 to 0.68) | 0.003 |
| Adjusted | 0.124–0.149† | | | |
| pDCD (pDCD vs No DCD) | | 1.056 | 0.348 (0.16 to 0.75) | 0.007 |
| Sex (men vs women) | | 0.173 | 0.841 (0.41 to 1.73) | 0.637 |
| Gestational age | | | | |
| 22–27 GW vs full-term | | 1.533 | 4.634 (0.56 to 38.23) | 0.154 |
| 28–31 GW vs full- term | | 0.716 | 2.046 (0.60 to 7.03) | 0.256 |
| 32–36 GW vs full-term | | 0.613 | 1.846 (0.73 to 4.66) | 0.194 |
| Neurodevelopmental disorder | | 0.806 | 0.447 (0.13 to 1.50) | 0.193 |
| Number of diagnoses posing a risk of motor problems | | 0.211 | 0.810 (0.64 to 1.02) | 0.074 |
| Number of hospital days | | 0.007 | 0.993 (0.98 to 1.01) | 0.467 |
| **USS graduation at 24 years** | | | | |
| Unadjusted | 0.045 | | | |
| pDCD (pDCD vs No DCD) | | 1.045 | 0.352 (0.13 to 0.96) | 0.041 |
| Adjusted | 0.089–0.102† | | | |
| DCD (DCD vs No DCD) | | 1.106 | 0.331 (0.11 to .96) | 0.041 |
| Sex (men vs women) | | 0.109 | 1.115 (0.39 to 3.15) | 0.838 |
| Gestational age | | | | |
| 22–27 GW vs full-term | | 19.464 | $284×10^6$ (0.00 to 0.00) | 0.999 |
| 28–31 GW vs full- term | | 0.271 | 0.763 (0.13 to 4.41) | 0.762 |
| 32–36 GW vs full-term | | 0.013 | 1.013 (0.26 to 3.99) | 0.985 |
| Neurodevelopmental disorder | | 0.195 | 0.823 (0.15 to 4.58) | 0.824 |
| Number of diagnoses posing a risk of motor problems | | 0.144 | 0.866 (0.63 to 1.19) | 0.371 |
| Number of hospital days | | 0.001 | 1.001 (0.98 to 1.03) | 0.925 |
| **USS graduation at 29 years** | | | | |
| Unadjusted | 0.040 | | | |
| pDCD (pDCD vs No DCD) | | 0.998 | 0.369 (0.13 to 1.06) | 0.063 |
| Adjusted | 0.079–0.090† | | | |
| pDCD (pDCD vs No DCD) | | 1.007 | 0.365 (0.12 to 1.11) | 0.076 |
| Sex (men vs women) | | 0.033 | 1.033 (0.34 to 3.11) | 0.953 |
| Gestational age | | | | |
| 22–27 GW vs full-term | | 18.979 | $175×10^6$ (0.00 to 0.00) | 0.999 |
| 28–31 GW vs full- term | | 0.365 | 0.694 (0.10 to 4.85) | 0.713 |
| 32–36 GW vs full-term | | 0.100 | 1.106 (0.26 to 4.89) | 0.892 |
| Neurodevelopmental disorder | | 0.407 | 0.666 (0.12 to 3.72) | 0.643 |
| Number of diagnoses posing a risk of motor problems | | 0.152 | 0.859 (0.62 to 1.19) | 0.360 |
| Number of hospital days | | 0.009 | 1.009 (0.98 to 1.04) | 0.579 |

*Reference.
†Range denotes the variance of the value of $R^2$ between different imputations.
DCD, developmental coordination disorder; GW, gestational weeks; pDCD, probable DCD; $R^2$, Neglekerke's $R^2$.

**Table 4** Educational outcomes for men and women with or without pDCD

| | Men n=108 | | | | Women n=75 | | | |
|---|---|---|---|---|---|---|---|---|
| | No DCD | pDCD | | | No DCD | pDCD | | |
| | n (%) | n (%) | | | n (%) | n (%) | | |
| | 80 (74.1) | 28 (25.9) | P value | g | 58 (77.3) | 17 (22.7) | P value | g |
| Age (years) at graduation from USS | | | | | | | | |
| Median | 19.0 | 19.0 | 0.157 | | 19.0 | 19.0 | 0.067 | |
| Min–max | 19.0–22.0 | 19.0–26.0 | | | 19.0–26.0 | 19.0–24.0 | | |
| IQR | 19.0–19.0 | 19.0–20.0 | | | 19.0–19.0 | 19.0–20.0 | | |
| Mean (SD) | 19.2 (.6) | 19.6 (1.4) | | 0.47 | 19.2 (1.0) | 19.8 (1.5) | | 0.85 |
| Missing data, total | 6 (7.5) | 4 (14.3) | | | 3 (5.2) | 5 (29.4) | | |
| No graduation | 6 (7.5) | 4 (14.3) | 0.281 | | 3 (5.2) | 3 (17.7) | 0.111 | |
| Absent/incomplete register data | | | | | | 2 (11.8) | | |
| USS graduation at 19 years | | | | | | | | |
| Yes | 60 (75.0) | 16 (57.1) | 0.075 | | 48 (82.8) | 8 (50.0) | 0.017 | |
| No | 20 (25.0) | 12 (42.9) | | | 10 (17.2) | 8 (50.0) | | |
| Missing data | | | | | | 1 (5.9) | | |
| USS graduation at 24 years | | | | | | | | |
| Yes | 74 (92.5) | 23 (82.1) | 0.148 | | 54 (93.1) | 13 (81.3) | 0.168 | |
| No | 6 (7.5) | 5 (17.9) | | | 4 (6.9) | 3 (18.8) | | |
| Missing data | | | | | | 1 (5.9) | | |
| USS graduation at 29 years | | | | | | | | |
| Yes | 74 (92.5) | 24 (85.7) | 0.281 | | 55 (94.8) | 13 (81.3) | 0.111 | |
| No | 6 (7.5) | 4 (14.3) | | | 3 (5.2) | 3 (18.8) | | |
| Missing data | | | | | | 1 (5.9) | | |
| Level of education at 28 years | | | | | | | | |
| Compulsory school, 9 years | 4 (5.1) | 2 (7.4) | 0.665 | | 3 (5.2) | 1 (7.1) | 0.082 | |
| Upper secondary education, ≤2 years | 2 (2.5) | 2 (7.4) | | | 0 (0) | 2 (14.3) | | |
| Upper secondary education, 3 years | 38 (48.1) | 14 (51.9) | | | 17 (29.3) | 5 (35.7) | | |
| Post-secondary education, <3 years | 14 (17.7) | 3 (11.1) | | | 10 (17.2) | 2 (14.3) | | |
| Bachelor's or master's degree | 21 (26.6) | 6 (22.2) | | | 28 (48.3) | 4 (28.6) | | |
| Missing data | 1 (1.3) | 1 (3.6) | | | | 3 (17.6) | | |

DCD, developmental coordination disorder; pDCD, probable DCD; USS, Upper Secondary School.

g=0.47). At age 28, fewer women with pDCD had attained a bachelor's or master's degree (n=4, 28.6%) compared with women without pDCD (n=28, 48.3%). Men with or without pDCD had comparable performance to women with pDCD (n=6, 22.2% vs n=21, 26.6%).

### Educational outcomes for pDCD and attention deficits

Half of the participants with pDCD had a co-occurring attention deficit according to CAMPB at 6.5 years of age (n=22, 48.9%) (table 5). Educational outcomes were similar between those with pDCD with or without a co-occurring attention deficit, but fewer of those with pDCD and an attention deficit had graduated from USS at all ages. The differences were most evident at 19 (42.9% vs 65.2%; p=0.137) and 24 (72.7% vs 90.9%; p=0.240) years of age. At the age of 28, the percentage of participants

with a degree from higher education was similar in both groups (23.8% vs 25.0%). However, among participants with pDCD and an attention deficit, 22.7% (n=5) had not graduated from USS by the age of 29.

### DISCUSSION

Using longitudinal individual level data from national registers on a cohort of NIC recipients, we found that their educational level was comparable to the national average at ages 19 and 24.[22] While fewer participants with pDCD had graduated from USS at these ages, the variation was similar between groups and most had graduated by age 24. By age 28, 33.1% of participants had attained a bachelor's or master's degree, similar to the national

**Table 5** Educational outcomes for participants with pDCD with (pDCD+) or without (pDCD-) a co-occurring attention deficit at 6.5 years of age

| | pDCD n (%) | pDCD– n (%) | pDCD+ n (%) | P value | d |
|---|---|---|---|---|---|
| | **45 (100)** | **23 (51.1)** | **22 (48.9)** | | |
| **Age (years) at graduation from USS** | | | | | |
| Median | 19.0 | 19.0 | 19.0 | 0.320 | |
| Min–max | 19.0–26.0 | 19.0–24.0 | 19.0–26.0 | | |
| IQR | 19.0–20.0 | 19.0–19.8 | 19.0–20.0 | | |
| Mean (SD) | 19.6 (1.4) | 19.5 (1.2) | 19.9 (1.8) | | 0.26 |
| Missing data, total | 9 (20.0) | 3 (13.0) | 6 (27.3) | | |
| No graduation | 7 (15.6) | 2 (8.7) | 5 (22.7) | 0.412 | |
| Absent/incomplete data | 2 (4.4) | 1 (4.3) | 1 (4.6) | | |
| **USS graduation at 19 years** | | | | | |
| Yes | 24 (54.5) | 15 (65.2) | 9 (42.9) | 0.137 | |
| No | 20 (45.5) | 8 (34.8) | 12 (57.1) | | |
| Missing data | 1 (2.2) | | 1 (4.5) | | |
| **USS graduation at 24 years** | | | | | |
| Yes | 36 (81.8) | 20 (90.9) | 16 (72.7) | 0.240 | |
| No | 8 (18.2) | 2 (9.1) | 6 (27.3) | | |
| Missing data | 1 (2.2) | 1 (4.3) | | | |
| **USS graduation at 29 years** | | | | | |
| Yes | 37 (84.1) | 20 (90.9) | 17 (77.3) | 0.412 | |
| No | 7 (15.9) | 2 (9.1) | 5 (22.7) | | |
| Missing data | 1 (2.2) | 1 (4.3) | | | |
| **Level of education at 28 years** | | | | | |
| Compulsory school, 9 years | 3 (7.3) | 1 (5.0) | 2 (9.5) | 0.755 | |
| Upper secondary education, ≤2 years | 4 (9.8) | 1 (5.0) | 3 (14.3) | | |
| Upper secondary education, 3 years | 19 (46.3) | 11 (55.0) | 8 (38.1) | | |
| Post-secondary education, <3 years | 5 (12.2) | 2 (10.0) | 3 (14.3) | | |
| Bachelor's or master's degree | 10 (24.4) | 5 (25.0) | 5 (23.8) | | |
| Missing data | 4 (8.9) | 3 (13.0) | 1 (4.5) | | |

DCD, developmental coordination disorder; pDCD, probable DCD; USS, Upper Secondary School.

average in Sweden.[24] We observed that fewer women with pDCD had graduated from USS at 19 years of age and more women without pDCD had attained a bachelor's or master's degree. However, apart from these findings, no other significant differences in educational outcomes were observed between sexes or among those with pDCD, with or without a co-occurring attention deficit according to CAMPB, at 6.5 years of age.

### DCD and educational outcomes

While previous research suggests that DCD can negatively affect educational outcomes in childhood and adolescence,[4–7] our findings indicate that this effect is less pronounced in adulthood. However, in our sample of NIC recipients, those with pDCD were less likely to graduate from USS at 19 and 24 years than their peers, and pDCD

predicted poorer educational outcomes independently of other factors affecting education, such as prematurity, sex and neurodevelopmental disorders.[14 22 25] Additionally, although the difference in post-secondary education completion rates was not statistically significant, those with pDCD (24.4%) were less likely to attain a bachelor's or master's degree by the age of 28 compared with those without DCD (35.8%). In Sweden, 30% of the general population (aged 25–64) attains a bachelor's degree on average at age 28.[3 24]

Our study suggests that adults with pDCD in childhood have positive educational outcomes based on graduation rates from USS. However, it is important to note that we did not consider academic performance or whether participants recieved any support, adjustments

and accommodations to successfully graduate. Previous research has shown that DCD can have a negative impact on examinations results and that students with motor difficulties may not receive adequate support in school.[5 40] Similarly, Yngve *et al* reported that despite the experienced need for support, half of students with special education needs did not receive accommodations in USS.[40] When interviewing participants from our cohort who had motor difficulties at 6.5 years of age, few reported remembering receiving any school-based support for their motor difficulties;[41] instead, they credited their parents for their success.

Despite being a relatively new diagnosis when our participants were in school, DCD remains unfamiliar to many.[42 43] According to parents, lack of teacher knowledge and awareness of DCD is the biggest challenge for their children with DCD in school,[44] with one in four parents reporting that their child did not enjoy going to school. Given that adolescents with DCD typically perform worse in school,[5] which is associated with higher rates of unemployment and lower wages,[3 20 21] raising awareness of DCD among educational professionals is essential for children with DCD to receive adequate support. Greater understanding and support from teachers and school staff can have a profound impact on the academic and long-term outcomes of individuals with DCD. The fact that a majority of our study participants with pDCD successfully completed USS and passed their examinations might be attributed to their participation in the initial cohort study. During this study, the children participated in repeated assessments throughout their childhood. This ongoing involvement may have helped parents gain a better understanding of their child's motor difficulties, how to support their child, and how to advocate for their child's needs. This, in turn, could potentially mitigate the adverse effects of DCD.

### Interaction between DCD and sex

Swedish women have considerably higher rates of graduation from USS and completion of post-secondary education than men.[22 25] When we investigated the educational outcomes within the sexes, we found that less women with pDCD had graduated from USS at 19 years of age compared with women without pDCD. In contrast, female NIC recipients without pDCD performed similar to or better than the general population.[22] A similar pattern was found in their educational attainment at age 28, where female NIC recipients without pDCD had achieved a bachelor's or master's degree to a greater extent than all other groups.

Our data suggest that DCD may impact men and women differently. While sex and gender differences in neurodevelopmental disorders are increasingly recognised,[45] this has not been extensively studied in relation to DCD. A recent study by Cleaton *et al* found that women with DCD experienced more gross-motor and non-motor difficulties as adults, and were more affected in their daily activities and participation, than men.[46] Additionally, while

children with DCD commonly perform worse academically than their typically developing peers,[5 8 10] De Waal *et al* found that boys and girls with DCD had different difficulties at school.[10] Factors related to sex and gender, as well as cultural beliefs, may further moderate and mediate the impact of neurodevelopmental disorders and DCD.[45 47] Also, girls with DCD appear to have lower self-competence appraisals compared with boys with DCD in athletic ability, physical appearance and academic competence,[18] and these differences become more pronounced with age. Although our findings are tentative, they contribute to the emerging understanding that DCD affects the sexes differently, which has implications for assessment and management of children with DCD, including possible support in school. Further research is needed to better understand these associations.

### Interaction between DCD and attention deficits

Co-occurring neurodevelopmental disorders such as DCD and ADHD can negatively impact various domains, including daily functioning and academic performance.[6 7 16] For example, a Swedish study reported that 36% of those diagnosed with ADHD and DCD at 6.6 years of age had not graduated from USS in their early thirties.[7] In our study, we observed that 22.7% of the participants with pDCD and an attention deficit at 6.5 years of age had not graduated from USS at age 29 compared with 9.1% of those with pDCD alone. Interestingly, we found no significant differences in educational outcomes between those with pDCD with or without a co-occurring attention deficit at 6.5 years of age. This may be because all children met the diagnostic criteria for DCD during childhood, indicating that their daily life was affected by their motor difficulties, and DCD in itself is known to affect educational achievements.[5 8 10] It is also worth noting that the study evaluated attention using CAMPB,[27 32] which only measures attention during the assessment and does not consider difficulties in other settings, and the children were not given an ADHD diagnosis. Additionally, the children's attention was dichotomised into having or not having an attention deficit without considering the type and severity of their attention deficit.

Another possible explanation for not finding differences between the groups is the symptomatic overlap between DCD and ADHD,[48 49] which can make it difficult to differentiate between the two disorders at a young age. Moreover, motor development is influenced by the individual, task and environment,[50 51] meaning that each individual's development is unique to their own abilities. Hence, supportive environments that provide opportunities for practice at an individual's level may have a positive effect on outcomes.[50 52] That the children in the initial cohort study participated in repeated assessments may have impacted the parents' knowledge and awareness about their child's difficulties. Additionally, they were guided in how to support and promote motor development and participation, which both may have positively affected outcomes. Furthermore, individuals

with DCD and ADHD learn to live with their disorders and develop strategies to overcome their difficulties.[41 53] Finally, all participants in the study are NIC recipients, and two-thirds were born premature (GW<37). Prematurity and low birth weight have been associated with neurodevelopmental disorders, including DCD and ADHD, as well as difficulties with executive functioning and inattention.[15 54] Addtionally, prematurity is associated with poorer cognitive and academic outcomes,[14 15 55 56] and an increased risk of not reaching curriculum goals during primary school.[11 56 57]

Research suggests that children with neurodevelopmental disorders, regardless of whether they were born prematurely or not, are at a higher risk of experiencing difficulties in various domains that are crucial for learning and academic performance. Therefore, any signs of any neurodevelopmental disorder should prompt a thorough evaluation of the child's abilities in all areas essential for their development and learning.[45] This will enable tailored interventions to optimise development, divert negative pathways, and mitigate negative outcomes.

## Strengths and limitations

While this study has several strengths such as its longitudinal design and use of national registers with high coverage and a low rate of missing data, there are also limitations that should be acknowledged. One potential issue is the small sample size, which may have limited the study's ability to detect true differences, especially in subgroup analyses with small group sizes. Therefore, caution should be taken when interpreting the results. Additionally, the study would have benefitted from a comparison to a population-matched control group and measures of the participants' functioning in adulthood to validate childhood assessments. Furthermore, we only measured educational outcomes as a binary outcome and did not consider grades or support received to graduate, making it difficult to assess the impact of motor difficulties on academic performance. Future research should investigate negative events such as lower grades or the resources required to graduate from USS. Regarding educational outcomes, we also lacked access to covariates that are known to affect education, such as socioeconomic background, parental education[17] and birth weight,[58] that could have informed our results. Additionally, neurodevelopmental disorders were not specified in the 'National Patient Registry' data, so we could not control for specific disorders. Moreover, the generalisability of our results may be limited by the characteristics of our study sample, as individuals that were healthy in their neonatal period may have other experiences and outcomes related to DCD. Furthermore, given the significant advances in neonatal care since the late 1980s, we do not know if this has affected outcomes. However, evidence suggests that increased survival rates among the extremely premature born infants has not been accompanied by decreased rates of neurodevelopmental disorders and educational difficulties,[15 59 60] making the results

relevant today. Finally, it is important to note that the age at follow-up may have affected the results, particularly in terms of the highest level of education, as study participants may continue to graduate later in life.

## CONCLUSIONS

This study suggests that pDCD during childhood may have a lasting impact on educational outcomes into adulthood, particularly among women. While further research is needed to determine whether these findings apply to the wider DCD population, they suggest that timely intervention and support for children with motor difficulties may play a significant role in improving learning and development. Raising awareness of DCD among parents, health and educational professionals is vital for early identification and the provision of appropriate support and interventions in schools, mitigating the potential negative consequences associated with DCD and promoting positive educational outcomes.

**Acknowledgements** We would like to acknowledge the children and parents who participated in the initial longitudinal cohort study. Additionally, we would like to thank Elvira Hemgren (PT), Kristina Persson (PT, PhD) and Bo Strömberg (MD, PhD) for their important work in collecting the baseline data, as well as Jill Zwicker (OT, PhD) and Amanda Kirby (MD, PhD) for their valuable advice during the initial phases of the project.

**Contributors** KJ acquired funding, conceptualised and designed the study with support from FS and RS. IP analysed and managed the data and drafted the initial manuscript. FS, TS, RS and KJ supported data analysis. KJ revised the manuscript. All authors critically reviewed the manuscript, contributed to drafts and approved the final version. We attest all authors meet ICMJE requirements for authorship. KJ acts as guarantor.

**Funding** This work was supported by the Gillbergska Foundation (grant/award no. n/a), Uppsala University Hospital through the Medical Training and Research Agreement Funds (ALF) (grant/award no. n/a); and Uppsala University through the Hedström Foundation and the Larsson Foundation (grant/award no. n/a).

**Competing interests** None declared.

**Patient and public involvement** Patients and/or the public were not involved in the design, or conduct, or reporting, or dissemination plans of this research.

**Patient consent for publication** Not applicable.

**Ethics approval** This study involves human participants and was approved by the Swedish Ethical Review Authority (reg. no. 2019-04119). The register data were delivered anonymised but linked to the participants via unique identification numbers administered by the Swedish Board of Health and Welfare. All data were stored at the Department of Women's and Children's Health at Uppsala University and on password protected servers held by the University. Data were handled in accordance to the General Data Protection Regulation and Swedish legislation. The processing of personal data was registered in Uppsala University's register of personal data processing and monitored by the Data Protection Officer. As the study is based on anonymised registry data from already collected data, we were granted permission to conduct the study without informing the participants and obtaining informed consent. The registers have been set up and are managed by public authorities according to current legislations. In order to access the data, both the National Board of Health and Welfare and Statistic Sweden required that the study was approved by the Swedish Ethical Review Authority. Before extracting the data, both authorities also conducted their own legal review.

**Provenance and peer review** Not commissioned; externally peer reviewed.

**Data availability statement** Data are available upon reasonable request. Data cannot be shared publicly because it includes potentially identifying and sensitive patient information, therefore access is limited to by request by the Swedish Ethical Review Authority (reg. no. 2019-04119). Data are available from Uppsala University,

Legal Affairs Division (contact via e-mail: registrator@uu.se), for researchers who meet the criteria for access to confidential data.

**ORCID iDs**
Filipa Sampaio http://orcid.org/0000-0002-5540-9853
Richard Ssegonja http://orcid.org/0000-0002-5323-5626
Kine Johansen http://orcid.org/0000-0002-9212-8452

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
