## [Reviewer comments · BMJ Open]

ARTICLE DETAILS

TITLE (PROVISIONAL)	The impact of developmental coordination disorder in childhood on educational outcomes in adulthood among neonatal intensive care recipients: a register-based longitudinal cohort study
AUTHORS	Persson, Isak; Sampaio, Filipa; Samkharadze, Tengiz; Ssegonja, Richard; Johansen, Kine

VERSION 1 – REVIEW

REVIEWER	Tal-Saban, Miri Hebrew University
REVIEW RETURNED	04-Feb-2023

GENERAL COMMENTS	The impact of DCD childhood on educational outcomes in adulthood: a cohort-based longitudinal register study This is an interesting article and important for the population of adults with DCD. The topic of this study can add knowledge about DCD . There are some issues that need to be in consideration: Introduction: 1. Overall, the introduction is clear and update but there is a need to add to the introduction sections about:• Academic abilities of DCD population,• CO-OCCURANDE of DCD and ADHD. Materials and methods: 1. The DCD population are only probable DCD because they were diagnosed only in their childhood and they didn't re- diagnose in adulthood. The authors need to explain why they didn't diagnose the participants, or at the very least, send a screening questionnaire (ADC or AACQ).2. Furthermore, it is problematic to diagnose the population as DCD because there were in neonatal care (criteria D in the DSM).3. Please add description of the assessment tools. In the data analysis / results: 4. There is a need to do the statistical analysis with ADHD as a co-variate5. Because the differences between the groups in gender there is a need to add statistical analysis of different between genders Discussion: 1. This section is not integrated and need to address the points above.
--

	2. This section is short in comparison to the other article sections. 3. The Limitation need to rewrite. Overall comments: 1. Figure 1 repeats what was said in the materials section 2. There are too many tables
--	---

REVIEWER	Smits-Engelsman, Bouwien Univ Cape Town
REVIEW RETURNED	12-Feb-2023

GENERAL COMMENTS	This paper investigates if Developmental Coordination Disorder (DCD) is related to poorer educational outcomes at childhood and adolescence and if they persist into adulthood. Longitudinal data are rare in DCD studies so this paper has valuable information, albeit for a special group of children with DCD (treated at NICU). As the authors mention their sample is not representative of the entire DCD-population. The paper is well written and the data warrant publication. The fact that NIC recipients had an educational level similar to that of the national average at 19 and 24 years of age is good news. In total, 33% of the participants had attained a bachelor's or a master's degree, and there was no statistically significant difference between the groups ($p = .080$); the regression model also shows the differences are small ($p = .063$). Children without DCD graduated at 19.2 years compared to 19.6 years among those with DCD. Median values for DCD and no DCD, female and male with DCD, DCD+ADHD are all 19 years for graduation USS (also the range is equal). You state "Our findings indicate that supporting these children through their academic career could help mitigate the impact of DCD on individuals and their families and reduce the economic burden of DCD on society". Haven't these children received help during their childhood and youth? Weren't they given extra time to finish exams etc? If there is an economic burden, how is this based on the data from this study? My main concern is the interpretation of the findings. I would conclude that main finding is that there are no (significant) differences and no meaningful differences between DCD and non-DCD NIC recipients. Which is a good message! If one looks at boys and girls separately there are no differences between boys with and without DCD and for the girls only USS graduation at 19 yrs is significant. The detailed analysis shows what causes this effect at age 19 (0.017). It is caused by the non-dcd girls to be very fast in graduation (83%) and DCD girls slower (50%). It is good to see that they catch up at age 24 when DCD girls 81% has graduated. It is important to notice that the effect (difference at 19) was caused by female NIC recipients without DCD who performed similar to, or better than, the general population. Meaning the DCD girls did not show very poor educational outcomes, but it looks that only difference in the detailed comparison is based on fact that that girls without DCD did really well. Of course there are extra problems that children with DCD encounter during their school carrier (like handwriting, self-esteem, lower sport participation, EF) and they definitely need support in school (as most children with neurodevelopmental disorders), but my overall impression of the data is that the boys are not different, ADHD does not lower the chances of finishing school in time. Girls are doing OK,
---

	comparable to DCD boys in outcome, while commonly girls do better in school than boys. Still girls with DCD have same or higher % of Bachelor's or master's degrees than boys in the end. The authors conclude "This study points towards DCD and DCD with a co-occurring attention deficit in childhood as being associated with poorer educational outcomes in NIC recipients in adulthood". My suggestion would be to turn this paper around: given good care NIC recipients with and without DCD, are similar (statistically not different) in most educational outcomes. If anything DCD girls are a bit slower than the comparison group (which has a larger range in graduation age?) Moreover for the 17 DCD girls in the data set, data seem to be missing for 5?. Because no differences between DCD and DCD +ADHD occurred, it seems that the explanation that cooccurring ADHD causes the educational problem is not true for this group. Minor Please make clear early in the paper that the "no motor deviations" and "motor delay "group were joined in a group "no DCD " Is there information why full-term (GW >37; n = 62, 33.9%) were in NIC? "Although it is difficult to draw any conclusion from this small high-risk sample, our findings are in line with previous research indicating that women with DCD are at an increased risk of experiencing adverse outcomes of their motor problems". Or difficult to draw any conclusion why NIC recipients without DCD (larger group) do better than the average girls in the national sample? Discuss the fact that no differences between DCD and DCD +ADHD occurred. The authors argue that the small sample size may have impacted on the study's ability to detect true differences, hence, the results need to be interpreted with caution. However, if you can't find difference in a group comparison of 138 and 45 children one should wonder about the clinical impact of a possible difference. Would an effect found in a larger group be clinically significant (is it possible to add effect sizes to the significant differences). Since no statistics values (U or Z so please add in tables) for the non-parametric testing is reported, I used the means to get an estimate of the effect size of the difference between the total groups; Cohen d 0.41.
--	---

VERSION 1 – AUTHOR RESPONSE

Reviewer 1	Introduction	
	Overall, the introduction is clear and update but there is a need to add to the introduction sections about:	We have reorganized, rewritten and added text about co-occurring neurodevelopmental disorders and academic abilities of the DCD population in the "Introduction" according reviewers request.

 • Academic abilities of DCD population, • co-occurrence of DCD and ADHD. 	
Materials and methods	
The DCD population are only probable DCD because they were diagnosed only in their childhood and they didn't re-diagnose in adulthood. The authors need to explain why they didn't diagnose the participants, or at the very least, send a screening questionnaire (ADC or AACQ).	We agree with reviewer 1 that a follow-up assessment in adulthood would be valuable. However, this is a register-based study based on anonymous data. The benefit of this study design is that all participants in the cohort are included, but at the same time we miss valuable information on their motor abilities in adulthood. We have conducted a questionnaire study where the participants have answered ADC (among others). The response rate was 54%. We are currently analysing these data. We have changed the term to probable DCD (pDCD) throughout the manuscript.
Furthermore, it is problematic to diagnose the population as DCD because there were in neonatal care (criteria D in the DSM).	Being born prematurely does not per se indicate a neurological disorder (criteria D), and children born prematurely can be given a diagnosis of DCD (see e.g. Spittle & Orton, 2014). The participants in the present study were assessed at repeated occasions from birth until 6.5 years of age, and those with other known neurological diagnosis such cerebral palsy were excluded from the study (see e.g. Montgomery et al., 2017). Please see Hemgren & Persson (2009) for a detailed description off the definitions of the motor deviation categories.
Please add description of the assessment tools.	We have added a description of the assessment methods under the section "Study design and participants".
Results	
There is a need to do the statistical analysis with ADHD as a co-variate.	We agree that this would have been very valuable, however, we do not have this information. In the National Patient Register, diagnoses were coded into groups of diagnoses (see supplementary file 3), but unfortunately, these are not specified in detail. Nevertheless, we have included a new variable called "Neurodevelopmental disorder" in our regression analysis (please see Table 3), using the information obtained from the register.

		Since we have no information regarding the specific type of neurodevelopmental disorder, and that an ADD or ADHD-diagnosis was not set within the project, we decided to investigate whether having a co-occurring attention deficit at 6.5 years of age assessed with CAMPB played a role in the participant with pDCD (please see Table 5). We have elaborated on this in the methods section.
	Because the differences between the groups in gender there is a need to add statistical analysis of different between genders.	We have included a table as a supplementary file showing the differences in educational outcomes between men and women in both the total sample and when comparing participants with and without pDCD (please see supplementary file 4). Moreover, we have added a sentence to address these results under the "Educational outcomes by sex" heading in the results section.
Discussion		
	This section is not integrated and need to address the points above.	We have rewritten and extended the discussion.
	This section is short in comparison to the other article sections.	We have rewritten and extended the discussion.
	The Limitation need to rewrite.	We have rewritten the strengths and limitations section including the bullet points.
Overall comments		
	Figure 1 repeats what was said in the materials section	We agree and we have moved Figure 1 to Supplementary file 1.
	There are too many tables	As we have moved Figure 1 to supplementary files, we are now within the limit of five figure and/or tables.
Reviewer 2	You state "Our findings indicate that supporting these children through their academic career could help mitigate the impact of DCD on individuals and their families and reduce the economic burden of DCD on society". Haven't these children received help during their childhood and youth? Weren't they given extra time to finish exams etc? If there is an economic burden, how is this	The children in our study were retrospectively classified into different motor deviation categories, including DCD, based on the DSM-TR-IV criteria (Hemgren & Persson, 2009). However, it is important to note that they did not receive a DCD-diagnosis at the time of assessment. Until the age of 6.5 years, they received standard treatment, which involved assessments, intervention targeting their motor difficulties, and parental guidance. Unfortunately, we do not have any information regarding interventions provided to the participants beyond this age, such as visits to physiotherapists

	based on the data from this study?	or special support in school through accommodations or adjustments. The limited utilization of the DCD diagnosis in Sweden, coupled with the frequent occurrence of adolescents with special education needs reporting insufficient school accommodations despite their perceived requirement for support (Yngve et al., 2019), suggests that it is not very likely that the participants to have received support for their motor difficulties specifically in school. In addition to the present study, we have interviewed a sample of participants from our cohort with motor difficulties at 6.5 years of age and found that few participants recalled receiving any school-based support for their motor difficulties, instead credited their success to their parents (Zahlander, 2020). We have removed the sentence about the economic burden and have adjusted our results, discussion, and conclusion accordingly. We also recommend that future studies investigate potential negative outcomes associated with academic achievements and DCD, such as the number of exams taken to pass and resource utilization.
	My main concern is the interpretation of the findings. I would conclude that main finding is that there are no (significant) differences and no meaningful differences between DCD and non-DCD NIC recipients. Which is a good message! If one looks at boys and girls separately there are no differences between boys with and without DCD and for the girls only USS graduation at 19 yrs is significant. The detailed analysis shows what causes this effect at age 19 (0.017). It is caused by the non-dcd girls to be very fast in graduation (83%) and DCD girls slower (50%). It is good to see that they catch up at age 24 when DCD girls 81% has graduated. It is important to notice that the effect (difference at 19) was caused by female NIC recipients without DCD who performed similar to, or better	We agree with the reviewer that it is positive that individuals with DCD seem to catch up to their peers, and we have modified the results and the discussion to address this comment. Example from the manuscript:  Educational outcomes – original Participants diagnosed with DCD at 6.5 years of age exhibited the lowest educational level on all variables (Table 2). There were statistically significant differences between groups for age at graduation from USS ($p = .002$) and the proportion of participants that had graduated at 19 ($p = .002$) and 24 ($p = .044$) years of age. In the whole sample, 16 participants had not finished USS (8.7%). Among these, seven were diagnosed with DCD (15.6%) compared to nine without DCD (6.5%). The mean age at graduation from USS for the whole sample was 19.3 years of age (± 1.0). There was a statistically significant difference between those diagnosed with and without DCD ($p = .022$), where those without DCD graduated at 19.2 years compared to 19.6 years among those with DCD. At 19 years of age, 78.3% of those without DCD had graduated from USS, compared to 54.5% among those with DCD ($p = .002$). At 24

	than, the general population. Meaning the DCD girls did not show very poor educational outcomes, but it looks that only difference in the detailed comparison is based on fact that that girls without DCD did really well. Of course, there are extra problems that children with DCD encounter during their school carrier (like handwriting, self-esteem, lower sport participation, EF) and they definitely need support in school (as most children with neuro-developmental disorders), but my overall impression of the data is that the boys are not different, ADHD does not lower the chances of finishing school in time. Girls are doing OK, comparable to DCD boys in outcome, while commonly girls do better in school than boys. Still girls with DCD have same or higher % of Bachelor's or master's degrees than boys in the end.	years of age, the corresponding numbers were 92.8% and 84.1% ($p = .044$), respectively. In total, 33% of the participants had attained a bachelor's or a master's degree, and there was no statistically significant difference between the groups ($p = .080$). However, 35.8% of those without DCD had attained a bachelor's or a master's degree at this age, while the corresponding number for those with DCD was 24.4%.  Educational outcomes – revised version The median age at USS graduation for all participants was 19 years (Table 2). While the pDCD group graduated with a slightly wider spread (IQR 19.0-20.0 vs 19.0-19.0) ($p = .022$, $g = 0.41$), the graduation age and range were similar between the groups. At 19 years, 78.2% ($n=108$) of the No DCD-group and had 54.5% ($n=24$) of the pDCD-group had graduated ($p = .002$). At 24 years, the percentage of graduates was higher in the No-DCD group (92.8%, $n=128$) than in the pDCD-group (81.1%, $n=36$) ($p = .044$). In total, 16 participants had not finished USS (8.7%) by the end of the study, seven of whom met the criteria for DCD (43.8%) at 6.5 years of age ($p = .069$). Regardless, the majority of participants (91.2%) had graduated from USS at 29 years. At 28 years, 33% of participants had attained a bachelor's or master's degree, with no statistically significant difference between the groups ($p = .080$). Nonetheless, the proportion of those without DCD with a bachelor's or master's degree (35.8%) was higher than those with pDCD (24.4%). Additionally, we have added effect sizes (see table 1, 2, 4 and 5). This analysis revealed that the difference in age at graduation from USS between women with and without pDCD has a large practical significance. It is important to note that the children did not receive an ADHD-diagnosis, but their attention deficit was assessed according to CAMPB (Hemgren & Persson, 1999). We had added information about CAMPB under "Study design and participants", as well as under "Measures". Additionally, we only included participants with pDCD in the sub-group analysis, which may have influenced the results. We elaborate on this in the discussion under the subheading "Interaction between DCD and attention deficits".
--	---	--

	The authors conclude “This study points towards DCD and DCD with a co-occurring attention deficit in childhood as being associated with poorer educational outcomes in NIC recipients in adulthood”. My suggestion would be to turn this paper around: given good care NIC recipients with and without DCD, are similar (statistically not different) in most educational outcomes. If anything, DCD girls are a bit slower than the comparison group (which has a larger range in graduation age?) Moreover, for the 17 DCD girls in the data set, data seem to be missing for 5? Because no differences between DCD and DCD +ADHD occurred, it seems that the explanation that cooccurring ADHD causes the educational problem is not true for this group.	We agree with the reviewer and have rewritten the results, discussion and conclusion to address this. Regarding our findings and ADHD, please see the comment above.
Minor		
	Please make clear early in the paper that the “no motor deviations” and “motor delay group” were joined in a group “no DCD”.	Thank you for the suggestion, and we agree that this is important information. We have added this under “Measures”.
	Is there information why full-term (GW >37; n = 62, 33.9%) were in NIC?	We have included the inclusion criteria for neonatal intensive care during the time of enrolment in the longitudinal cohort study under “Study design and participants”.
	“Although it is difficult to draw any conclusion from this small high-risk sample, our findings are in line with previous research indicating that women with DCD are at an increased risk of experiencing adverse outcomes of their motor problems”. Or difficult to draw any conclusion why NIC recipients without DCD (larger group) do better than the	The variation in education levels within Sweden may explain these gender differences. National statistics indicate that highly educated individuals are mainly concentrated in metropolitan areas and municipalities with universities (Statistic Sweden, 2019). Women generally have higher levels of education compared to men. More than a quarter of the population is highly educated, with a significant presence in metropolitan areas. Stockholm County stands out in terms of education, with at least a third of the population in nine municipalities having completed three or more years of post-secondary education. Uppsala is

	average girls in the national sample?	another municipality with a notable level of education. The majority of the participants in our study reside in Stockholm and Uppsala. Among women aged 25-44, over half have pursued post-secondary education, and 40 percent of them are highly educated. In contrast, a quarter of men in the same age group have attained a high level of education. This gender difference can be attributed to a higher number of women pursuing higher education. Currently, women constitute 61 percent of registered students in Swedish universities and colleges. We have rewritten the discussion and elaborated on potential reasons for the differences between men and women in the discussion and the conclusion.
	Discuss the fact that no differences between DCD and DCD +ADHD occurred.	We have elaborated on this in the discussion under the subheading "Interaction between DCD and attention deficits".
	The authors argue that the small sample size may have impacted on the study's ability to detect true differences, hence, the results need to be interpreted with caution. However, if you can't find difference in a group comparison of 138 and 45 children one should wonder about the clinical impact of a possible difference. Would an effect found in a larger group be clinically significant (is it possible to add effect sizes to the significant differences). Since no statistics values (U or Z so please add in tables) for the non-parametric testing is reported, I used the means to get an estimate of the effect size of the difference between the total groups; Cohen d 0.41.	We have added effect sizes were appropriate. Effect sizes were calculated using Hedges g and Cohen's d (Kumar et al., 2022)